# Molecular Mechanisms of Radiation Resistance in Breast Cancer: A Systematic Review of Radiosensitization Strategies

**DOI:** 10.3390/cimb47080589

**Published:** 2025-07-24

**Authors:** Emma Mageau, Ronan Derbowka, Noah Dickinson, Natalie Lefort, A. Thomas Kovala, Douglas R. Boreham, T. C. Tai, Christopher Thome, Sujeenthar Tharmalingam

**Affiliations:** 1School of Natural Sciences, Laurentian University, Sudbury, ON P3E 2C6, Canada; emageau@laurentian.ca (E.M.); ndickinson@laurentian.ca (N.D.);; 2Undergraduate Medical Education, NOSM University, 935 Ramsey Lake Rd., Sudbury, ON P3E 2C6, Canada; nlefort@nosm.ca; 3Medical Sciences Division, NOSM University, 935 Ramsey Lake Rd., Sudbury, ON P3E 2C6, Canada; 4Health Sciences North Research Institute, Sudbury, ON P3E 2H2, Canada

**Keywords:** breast cancer, radiation resistance, radiosensitization, PI3K/AKT/mTOR, DNA damage repair, non-coding RNA, microRNA, cancer stem cells, CD44^+^/CD24^−^/low, tumor microenvironment, epithelial–mesenchymal transition, immune modulation, lncRNA, therapeutic targets, non-homologous end joining

## Abstract

Breast cancer remains one of the most prevalent malignancies worldwide, and radiation therapy is a central component of its management. However, intrinsic or acquired resistance to radiation significantly compromises therapeutic efficacy. This systematic review aimed to identify and evaluate molecular mechanisms and interventions that influence radiation sensitivity in breast cancer models. A comprehensive PubMed search was conducted using the terms “breast cancer” and “radiation resistance” for studies published between 2002 and 2024. Seventy-nine eligible studies were included. The most frequently investigated mechanisms included the dysregulation of the PI3K/AKT/mTOR and MAPK signaling pathways, enhanced DNA damage repair via non-homologous end joining (NHEJ), and the overexpression of cancer stem cell markers such as CD44^+^/CD24^−^/low and ALDH1. Several studies highlighted the role of non-coding RNAs, particularly the lncRNA DUXAP8 and microRNAs such as miR-21, miR-144, miR-33a, and miR-634, in modulating radiation response. Components of the tumor microenvironment, including cancer-associated fibroblasts and immune regulators, also contributed to radiation resistance. By synthesizing current evidence, this review provides a consolidated resource to guide future mechanistic studies and therapeutic development. This review highlights promising molecular targets and emerging strategies to enhance radiosensitivity and offers a foundation for translational research aimed at improving outcomes in radiation-refractory breast cancer.

## 1. Introduction

Breast cancer is the most commonly diagnosed cancer in women worldwide, excluding non-melanoma skin cancers [1]. In the United States, the incidence of invasive breast cancer has continued to rise, with an estimated 287,850 new cases and 43,250 deaths in 2022 [2]. The number of new breast cancer cases has risen even further since then, with an estimated 310,720 new cases of invasive breast cancer and 42,250 deaths having occurred in 2024 in the United States alone [3]. Globally, 2.3 million new breast cancer cases were diagnosed in 2022, underscoring the urgent need for continued research into effective prevention and treatment strategies [4].

Breast cancer can arise from various tissues within the breast, leading to multiple histological subtypes. These include, but are not limited to, ductal, lobular, mixed tumors, mucinous, and inflammatory breast cancer (Figure 1) [5,6]. Ductal breast cancer has the highest prevalence, accounting for between 50 and 80% of all breast cancer cases [6]. It is formed in the ductal region of the breast, which transports milk from the lobules to the nipples [7]. Similarly, lobular breast cancer forms in the milk producing lobules attached to the ducts [7]. This subtype accounts for approximately 5–15% of cases [6,8]. Mixed tumors are characteristic of cancer growing in both the ducts and lobules and accounts for approximately 25.3% of all breast cancer cases [6]. Mucinous breast cancer demonstrates a greater presence of extracellular mucin and affects around 2% of those diagnosed with breast cancer [6,9]. Lastly, inflammatory breast cancer presents as inflammation, redness, or changes to the skin of the breast and is a more rare and aggressive type of breast cancer, with an incidence rate of roughly 2–4% [5].

Triple-negative breast cancer (TNBC) is an aggressive subtype of breast cancer characterized by the absence of estrogen receptors (ER), progesterone receptors (PR), and human epidermal growth factor receptor 2 (HER2) [10]. Due to this lack of targetable receptors, TNBC does not respond to hormone therapies or HER2-targeted treatments, limiting therapeutic options [11]. TNBC most commonly affects premenopausal women, typically under the age of 40, and is associated with a high rate of invasion and metastasis [12]. Approximately 40% of patients diagnosed with TNBC between stages I-III will develop a recurrence of the disease in the years following the initial diagnosis [13]. This highlights the urgent need for novel therapeutic strategies.

Breast cancer is classified into stages based on tumor size, lymph node involvement, and the extent of metastasis, data which collectively guide treatment planning and prognosis [7]. The disease is staged from 0 to IV, with increasing severity [14]. Stage 0, or carcinoma in situ, represents the earliest form and is characterized by the presence of abnormal cells confined to the ducts or lobules without invasion into surrounding tissue [7,15]. In contrast, Stage IV, or metastatic breast cancer, is the most advanced stage, marked by the spread of cancer cells beyond the breast and regional lymph nodes to distant organs such as the bones, lungs, liver, or brain [16].

Treatment options for breast cancer vary widely and depend on both the cancer subtype and stage at diagnosis. Common therapeutic approaches include surgery, radiation therapy, chemotherapy, hormonal therapy, and targeted therapy, often used in combination to maximize efficacy [15]. Among these, radiation therapy remains a cornerstone of breast cancer [17]. It is commonly delivered via external beam radiation and can be used in adjuvant (post-surgical), neoadjuvant (pre-surgical or post-mastectomy recurrence), or palliative (symptom relief in metastatic disease) settings [17]. Despite its clinical benefits, repeated radiation exposure can paradoxically promote adaptive responses in cancer cells, ultimately leading to radiation resistance [18]. Unfortunately, breast cancer frequently develops resistance to radiation therapy, increasing the risk of post-treatment recurrence and resulting in a more challenging form of the disease to manage effectively [19].

Radiation therapy employs ionizing radiation to disrupt the genetic material of cancer cells, thereby inhibiting cell division and promoting apoptosis [20]. Ionizing radiation, such as X-rays and gamma rays, penetrates tissues and induces DNA damage by breaking chemical bonds and ionizing atoms [21]. This damage may occur through direct interactions with DNA or indirectly via the generation of reactive oxygen species (ROS) that subsequently damage genetic material (Figure 2). While this is lethal to many cancer cells, a subset of cells may survive and acquire resistance, posing significant challenges for treatment efficacy and long-term disease control [22].

Radiation resistance manifests in many ways but is often associated with an epithelial to mesenchymal transition (EMT), mutation to tumor suppressor genes, and enhanced DNA damage repair [23,24,25]. EMT involves the conversion of epithelial cells into a mesenchymal phenotype, conferring enhanced motility, invasiveness, and resistance to radiation-induced stress [25]. This phenotypic shift is associated with greater metastatic potential and tumor progression [26].

Tumor suppressor genes, such as TP53, play critical roles in regulating cell growth and apoptosis to prevent tumorigenesis [27]. However, mutations or inactivation of these genes are common in cancer and contribute to uncontrolled cell proliferation and therapy resistance [23]. Additionally, the DNA damage response (DDR) is a key cellular mechanism that detects and repairs DNA damage [24]. In cancer cells, the upregulation of DNA repair pathways enables survival following radiation therapy, thereby facilitating resistance and continued tumor growth despite treatment [28].

Radiation resistance remains a critical challenge in the management of breast cancer, often contributing to treatment failure and disease recurrence. Although numerous studies have investigated the molecular mechanisms underlying resistance, findings are often fragmented across distinct pathways and targets, with limited cross-comparison or integration. This systematic review aims to summarize and synthesize mechanistic studies that explore the development of radiation resistance in breast cancer, encompassing signaling cascades, DNA repair pathways, epigenetic regulators, and immune/stromal contributors. By consolidating these diverse lines of evidence, this review highlights key radiosensitization targets, identifies gaps in translational relevance, and provides a foundation for future studies to advance therapeutic strategies and improve radiotherapy outcomes.

## 2. Methodology

This systematic review was conducted in accordance with the PRISMA (Preferred Reporting Items for Systematic Reviews and Meta-Analyses) guidelines. A comprehensive literature search was performed using the PubMed database to identify relevant studies published between 2002 and 2024. The advanced search function was used, with the keywords “breast cancer” AND “radiation resistance” applied to both the title and abstract fields. To refine the results, review articles were excluded by applying the filter “Review [Publication Type]” in combination with the search term NOT. Additional exclusion criteria included studies published before 2002, those using non-mammalian models, articles focusing on non-breast cancers, and studies that did not investigate a specific molecular target related to radiation resistance. A total of 135 articles were identified initially. After applying the inclusion and exclusion criteria, 79 full-text articles were retained for analysis. The study selection process is summarized in Figure 3.

To synthesize the findings, the included articles were categorized into mechanistic themes based on their focus, including cancer stem cells, DNA repair pathways, cell signaling, microRNAs, and emerging molecular targets. Relevant data from each study were extracted and compiled into summary tables, which present: the authors, year of publication, PubMed ID, cell lines or models used, molecular target or strategy investigated, and whether the target was found to increase or decrease radiation resistance.

## 3. Results and Discussion

### 3.1. The Role of Cancer Stem Cells in Breast Cancer Radiation Resistance

Cancer stem cells (CSCs) are known to be radioresistant and share key features with normal stem cells, including the ability to self-renew and differentiate [17]. Relevant studies investigating CSC markers and their association with radiation resistance in breast cancer are summarized in Table 1. Among the most frequently studied markers are the surface receptors CD44 and CD24. Breast CSCs often exhibit a CD44-positive, CD24-negative or low (CD44^+^/CD24^−^/low) phenotype, which has been repeatedly linked to radioresistance [29]. Several studies have demonstrated that radiosensitivity in CD44^+^/CD24^−^/low cells can be enhanced by targeting downstream molecules such as ALDH1, p-S6K1, ATM, MAPK1, and IFIT2 [29,30,31,32,33]. Supporting the clinical relevance of this phenotype, a cohort study of 61 Indian women aged 25–75 found that among CD44^+^/CD24^−^/low breast cancer cells, 82.5% lacked estrogen receptors (ER), 85% lacked progesterone receptors (PR), and 90% were HER2-positive [34]. Interestingly, when stratified by hormone receptor status or HER2 expression, the frequency of CD44^+^/CD24^−^/low expression remained nearly identical [34]. In TNBC specifically, 76.9% of tumors expressed the CD44^+^/CD24^−^/low phenotype, suggesting its strong association with radioresistance in this aggressive subtype [34].

The CD44^+^/CD24^−^/low phenotype is often associated with elevated expression of aldehyde dehydrogenase 1 (ALDH1), an enzyme involved in aldehyde detoxification, cellular protection, and stem cell differentiation [31,32]. In vitro studies using CD44^+^/ALDH1^high^ MDA-MB-231 cells showed significantly greater survival and colony-forming ability following treatment with standard chemotherapeutics (paclitaxel and doxorubicin) and radiation, compared to CD44^−^/ALDH1^low^ counterparts [31]. Notably, radiation sensitivity was enhanced when ALDH1 levels were reduced using all-trans retinoic acid (ATRA) or inhibited using diethylaminobenzaldehyde, a selective ALDH1 inhibitor [31]. Diethylaminobenzaldehyde not only suppressed growth in CD44^+^/ALDH1^high^ cells but also sustained radiosensitization over time, suggesting that direct inhibition of ALDH1 enzymatic activity may be a promising strategy to overcome radiation resistance in CSC-enriched breast cancer populations [31].

The mammalian target of rapamycin (mTOR) signaling pathway has also been implicated in radiation resistance among breast cancer stem cell populations [30]. In particular, phosphorylated ribosomal S6 kinase 1 (p-S6K1), a downstream effector of mTOR, was found to be overexpressed in CD44^+^/CD24^−^/low MCF7 cells compared to parental controls, suggesting a role in therapy resistance [30]. Targeting this axis with everolimus, an FDA-approved mTOR inhibitor, in combination with radiation, significantly increased radiosensitivity in CD44^+^/CD24^−^/low MCF7 cells [18]. Notably, everolimus alone did not induce cell death prior to irradiation; however, when administered pre-irradiation, it markedly enhanced radiation-induced cytotoxicity, supporting its potential as a radiosensitizing agent in CSC-enriched tumors [30].

The ataxia-telangiectasia mutated (ATM) signaling pathway, a key regulator of the DNA damage response, was found to be upregulated in CD44^+^/CD24^−^/low MDA-MB-231 and MCF-7 cells [33]. Similarly to the approach employed by Choi et al. [30], the researchers used a targeted inhibitor to modulate signaling activity—in this case, KU55933, a selective ATM inhibitor (2-Morpholin-4-yl-6-thianthren-1-yl-pyran-4-one) [33]. Treatment with KU55933 resulted in a tenfold increase in radiation sensitivity in the CSC-enriched population compared to untreated controls, reinforcing the potential of targeting DNA repair pathways to overcome radiation resistance in CD44^+^/CD24^−^/low breast cancer cells [33].

A different strategy was employed by Koh et al. [29], who focused on interferon-induced protein with tetratricopeptide repeats 2 (IFIT2), a protein implicated in metastasis and recurrence. Using MDA-MB-231 cells with the CD44^+^/CD24^−^/low phenotype, the authors induced radiation resistance through repeated irradiation—25 cycles of 2 Gy over 5 weeks—to mimic acquired resistance [29]. To counter this, they tested baicalein, a naturally occurring flavonoid with known anti-inflammatory properties, as a radiosensitizing agent. Baicalein treatment effectively downregulated IFIT2 expression and significantly reduced CD44^+^/CD24^−^/low marker expression, suggesting a loss of stem-like traits and reduced metastatic potential in resistant cells [29]. Additionally, baicalein induced dose-dependent apoptosis, highlighting its dual role as both a chemotherapeutic and radiosensitizing compound in the treatment of radiation-resistant breast cancer [29].

In addition to the well-characterized CD44^+^/CD24^−^/low phenotype, several studies have identified other CSC-associated markers and signaling regulators contributing to radiation resistance. For example, epithelial cell adhesion molecule (EpCAM) was shown to be elevated in ZR-75-1 and MCF-7 cells and linked to increased radioresistance, suggesting its role as a potential CSC marker [35]. Similarly, Lin28, a stemness-associated RNA-binding protein, was found to promote radiation resistance across multiple ER^+^ breast cancer cell lines including T47D and MCF-7 [36]. Other targets such as neuropilin-1 (NRP1) [37] and glucose-regulated protein 78 (GRP78) [38] have also been associated with therapy-resistant phenotypes, implicating developmental and stress-response pathways in CSC-mediated radiation resistance. These findings underscore the heterogeneity of CSC markers and highlight alternative molecular targets—beyond CD44 and ALDH1—that may be exploited to overcome resistance in breast cancer.

Overall, these studies highlight the key role of CD44^+^/CD24^−^/low breast cancer stem cells in mediating radiation resistance via pathways involving ALDH1, p-S6K1, ATM, and IFIT2. Additional CSC-associated markers such as EpCAM, Lin28, NRP1, and GRP78 further underscore the molecular heterogeneity of resistance mechanisms. Figure 4 summarizes these stem cell–associated targets and their roles in promoting or mitigating radiation resistance.

**Table 1 cimb-47-00589-t001:** Summary of studies investigating cancer stem cell markers in breast cancer radiation resistance. This table summarizes key studies examining the role of cancer stem cell-associated markers or strategies in modulating radiation resistance in breast cancer. For each study, the authors, year of publication, and PubMed ID (PMID) are provided to identify the source. The model systems used (human breast cancer cell lines or xenografts), molecular targets or interventions, and their observed effect on radiation resistance—either an increase (I) or decrease (D)—are listed. In cases where a defined molecular target was not specified, the experimental strategy or cellular condition investigated is noted.

Authors, Year of Publication	PMID	Model System	Molecular Target/ Intervention	Effect on Radiation Resistance [Increase (I)/Decrease (D)]
Anand et al., 2023 [34]	36891450	CD44^+^/CD24^−^ breast cancer biopsies	CD44^+^/CD24^−^	I
Bensimon et al., 2016 [39]	25641732	MCF-7, MCF-7-CD24^low^, MCF-7-CD24^neg^, MDA-MB-436 and MDA-MB-436-CD24^high^	CD24	D
Bensimon et al., 2013 [40]	22330142	T-47D, BT-20, MDA-MB-157, MDA-MB-231	CD24(−/low)	I
Bontemps et al., 2022 [41]	36367190	MCF-7, MCF-7_CD24−, T47D, T47D_CD24−, HMLE	CD24(−/low)	I
Choi et al., 2020 [30]	31959810	MCF7 (CD44^high^/CD24^low^), T47D, ZR-751, BT474, SKBR3, MDA-MB-453, MDA-MB-231	p-S6K1	I
Croker & Allan, 2012 [31]	21818590	MDA-MB-231, MDA-MB-468 (ALDH^hi^CD44+/ALDH^low^CD44−)	ALDH	I
Inalegwu et al., 2022 [42]	35579852	MCF-7, Fractionally irradiated cells (FIR20)	↑ Stemness	I
Koh et al., 2019 [29]	30875792	MDA-MB-231, MDA-MB-231/IR	IFIT2	I
Li et al., 2013 [38]	24002052	MCF-7	GRP78	I
Mal et al., 2021 [35]	33490064	ZR-75-1, ZR-75-1^EpCAM^, ZR-75-1^FR^ and MCF-7, MCF-7^FR^	EpCAM	I
Sabol et al., 2020 [43]	32326381	MCF-7, ZR-75, T47D	Obesity-altered adipose stem cells	I
Wang et al., 2013 [36]	23840685	T47D, MCF-7, Bcap-37, SK-BR-3	Lin28	I
Wang et al., 2023 [37]	36333630	SK-BR-3, MDA-MB-468, MDA-MB-231, MCF-7, MCF10A	NRP1	I
Wei et al., 2011 [44]	22023707	AS-B145 (ALDH+), AS-B244 (ALDH+), 4T1 (ALDH+), MDA-MB-231 (ALDH+)	Hsp27	I
Woodward et al., 2007 [45]	17202265	MCF-7	Wnt/β-catenin pathway	I
Yan et al., 2016 [46]	27036550	MCF-7/C6 (CD44(+)/CD24(−/low))	ATRA	D
Yin & Glass, 2011 [33]	21935375	CD44(+)/CD24(− or low) subset of MCF-7, MDA-MB-231, MDA-MB-436, BD20, HCC38, HCC1937	ATM signaling	I
Zielske et al., 2011 [47]	21804918	MC1, UM2, patient-derived xenografts	CD44(+) CD24(−) lin(−)	D

Abbreviations: CD24 (Cluster of Differentiation 24), p-S6K1 (phosphorylated ribosomal S6 kinase), ALDH (aldehyde dehydrogenase), IFIT2 (Interferon-Induced Protein with Tetratricopeptide Repeats 2), GRP78 (glucose-regulated protein 78), EpCAM (epithelial cell adhesion molecule), NRP1 (Neuropilin-1), Hsp27 (Heat Shock Protein 27), ATRA (all-trans retinoic acid), ATM (ataxia-telangiectasia mutated), CD44 (Cluster of Differentiation 44).

### 3.2. DNA Repair and Redox Pathways as Determinants of Radiation Resistance

Exposure to ionizing radiation induces DNA damage, making DNA repair pathways a central consideration in understanding mechanisms of radiation resistance. Relevant studies investigating DNA damage response and repair pathways in breast cancer models are summarized in Table 2. In one such study, Wang et al. [48] examined the role of flap endonuclease 1 (FEN1) in MDA-MB-231 cells. FEN1 is a key enzyme in DNA replication and repair and is often upregulated in response to chemotherapeutic agents and ionizing radiation, enabling cancer cells to survive and proliferate despite genotoxic stress. The authors also investigated the role of YY1, a transcriptional repressor known to bind the FEN1 promoter. Under normal conditions, YY1 suppresses FEN1 expression; however, radiation or chemotherapy exposure was found to reduce YY1 levels, leading to increased FEN1 expression and enhanced cellular repair capacity [48]. Restoration of YY1 expression repressed FEN1 and increased radiosensitivity, suggesting that the YY1–FEN1 axis may represent a viable therapeutic target to overcome radiation resistance in breast cancer (Figure 5) [48].

Another DNA repair mechanism frequently implicated in radiation resistance is the non-homologous end joining (NHEJ) pathway, which is responsible for repairing the majority of cellular DNA double-strand breaks. Tian et al. [49] investigated the role of RUVBL1 (RuvB-like AAA ATPase 1), a protein ubiquitinated by DTL (Denticleless E3 Ubiquitin Protein Ligase Homolog), leading to the formation of a RUVBL1/2–β-catenin complex. This complex was shown to enhance NHEJ activity and promote cell survival following radiation exposure. Similarly, Andrade et al. [50] explored the function of ARID1A, a tumor suppressor that also facilitates NHEJ. They found that HuR (also known as ELAVL1, Embryonic Lethal Abnormal Vision-Like 1), an RNA-binding protein, stabilizes ARID1A (AT-Rich Interaction Domain 1A) mRNA in TNBC cells, leading to increased ARID1A expression and heightened resistance to radiation. Building on this, Mehta et al. [51] demonstrated that silencing HuR in TNBC cells disrupted redox homeostasis, resulting in increased reactive oxygen species (ROS) accumulation, enhanced DNA damage, and improved radiation sensitivity. These studies are collectively illustrated in Figure 6, and highlight the importance of NHEJ regulation and redox balance in mediating radiation resistance, particularly in aggressive breast cancer subtypes.

**Figure 6 cimb-47-00589-f006:**
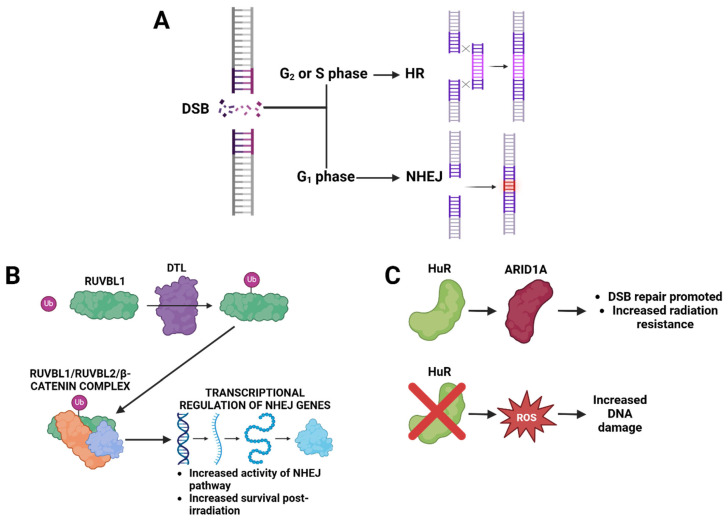
DNA double-strand break (DSB) repair mechanisms and their role in radiation resistance. (**A**) DSBs can be repaired through homologous recombination (HR) when a sister chromatid is available as a template, typically during the S and G2 phases of the cell cycle [52]. Alternatively, non-homologous end joining (NHEJ) directly ligates the broken DNA ends without a homologous template [52]. (**B**) Denticleless E3 ubiquitin protein ligase homolog (DTL) ubiquitinates RuvB-like AAA ATPase 1 (RUVBL1) to facilitate the formation of the RUVBL1/2–β-catenin complex, which transcriptionally upregulates genes involved in the NHEJ pathway, thereby enhancing DNA repair and promoting radiation resistance (**C**) Human antigen R (HuR), an RNA-binding protein often overexpressed in breast cancer, stabilizes AT-rich interaction domain 1A (ARID1A) mRNA, leading to increased ARID1A expression and enhanced DSB repair [39]. Knockdown of HuR reduces ARID1A levels, increases reactive oxygen species (ROS), and sensitizes cells to radiation by impairing DNA repair. Created in BioRender. Dickinson, N. (2025) https://BioRender.com/jhixrx1.

Some studies have explored therapeutic strategies to disrupt DNA repair pathways and overcome radiation resistance. Wang et al. [53] investigated the role of fatty acid synthase (FASN), which is frequently overexpressed in breast cancer and linked to enhanced DNA repair and resistance to radiation. Although no FDA-approved FASN inhibitors currently exist due to poor bioavailability and toxicity concerns, the study found that proton pump inhibitors (PPIs)—specifically R-enantiomers like dexlansoprazole—showed promise in inhibiting FASN activity and sensitizing cells to radiation [53]. In a separate approach, Nashir Udden et al. [54] examined the interplay between endocrine therapy resistance and radiation resistance in T47-D and MCF-7 cells. ER^+^ breast cancers are often treated with endocrine therapy prior to surgery; however, resistance can emerge due to mutations in *ESR1*, the gene encoding the estrogen receptor [54]. These mutations were also associated with radiation resistance, partly mediated through upregulation of BRD4, a member of the bromodomain and extraterminal (BET) protein family involved in DNA repair. Treatment with the BET inhibitor OTX015 sensitized ESR1-mutant cells to radiation, suggesting that BET inhibition may serve as a dual-targeting strategy in endocrine- and radiation-resistant breast cancers [54].

In addition to the canonical DNA repair mechanisms already discussed, several other studies identified complementary pathways and strategies that contribute to radiation resistance. Sencan et al. [55] investigated UVRAG (UV radiation resistance–associated gene), a regulator of both autophagy and DNA repair, and demonstrated that its upregulation enhances radiation resistance across multiple breast cancer cell lines. These findings suggest that stress response pathways, such as autophagy, may intersect with DNA repair to promote cell survival following irradiation.

Several studies also highlighted the importance of alternative end joining pathways in sustaining radiation resistance. Kumar et al. [56] and Lamb et al. [57] demonstrated that DNA-PK– and polymerase theta (Pol θ)–dependent repair mechanisms contribute to cellular recovery from DNA double-strand breaks, particularly when homologous recombination is impaired. This underscores the therapeutic potential of targeting backup repair mechanisms in radioresistant tumors.

Lastly, epigenetic modulation also shows promise for radiosensitization. Chiu et al. [58] used histone deacetylase inhibitors (HDACi) to induce misfolded protein accumulation, thereby disrupting proteostasis and reducing radiation resistance in breast cancer models.

In parallel, oxidative stress regulation has emerged as another critical determinant of radiation response. Wang et al. [59] demonstrated that a Cu-doped polypyrrole-based hydrogel, when injected intratumorally, suppressed antioxidant capacity and increased ROS production, thereby enhancing radiation sensitivity both in vitro and in vivo. This finding complements previous studies targeting HuR and ARID1A, and highlights ROS amplification as a viable therapeutic strategy. Additional studies reinforce the importance of redox homeostasis in maintaining radiation resistance. Wu et al. [60] found that Bcl-2, a mitochondrial membrane protein with anti-apoptotic function, was upregulated in breast cancer cells and protected against radiation-induced apoptosis by limiting ROS accumulation. Similarly, Abdullah et al. [61] showed that inhibition of thioredoxin reductase, a key antioxidant enzyme, increased intracellular ROS levels and sensitized breast cancer cells to radiation. Zhou et al. [62] reported that SOD2 and CDKN1A were upregulated in MDA-MB-231 cells, contributing to improved oxidative stress tolerance and radiation survival. Finally, Diaz et al. [63] identified peroxiredoxin II as another ROS-scavenging enzyme upregulated in MCF-7 cells, further supporting the role of enhanced antioxidant defenses in sustaining radiation resistance.

Together, these studies highlight the multifaceted nature of radiation resistance in breast cancer, driven by both DNA repair fidelity and oxidative stress regulation. While NHEJ and HR remain central repair mechanisms, alternative end joining, autophagy, and redox homeostasis also contribute to cellular survival. Targeting these pathways—using agents such as BET and HDAC inhibitors, ROS amplifiers, and metabolic modulators—represents a promising strategy to enhance radiosensitivity and improve therapeutic outcomes.

**Table 2 cimb-47-00589-t002:** Summary of studies investigating DNA repair pathways in breast cancer radiation resistance. This table presents studies that examine molecular targets and interventions affecting radiation resistance through modulation of DNA repair, genome stability, or oxidative stress response pathways in breast cancer models. For each study, the authors, year of publication, and PubMed ID (PMID) are provided to identify the source. The model systems used (human breast cancer cell lines or xenografts), molecular targets or interventions, and their observed effect on radiation resistance—either an increase (I) or decrease (D)—are listed. In cases where a defined molecular target was not specified, the experimental strategy or cellular condition investigated is noted.

Authors, Year of Publication	PMID	Model System	Molecular Target/ Intervention	Effect on Radiation Resistance [Increase (I)/Decrease (D)]
Abdullah et al., 2021 [61]	33768386	MDA-MB-231, MDA-MB-468, MDA-MB-436, MCF-7, T47D,	Thioredoxin reductase	I
Andrade et al., 2019 [50]	31847141	MDA-MB-231, Hs578t, MCF-7, MDA-MB-468	HuR and ARID1A	I
Barlow et al., 2024 [64]	38467328	MDA-MB-468, MDA-MB-231	FASN	I
Chiu et al., 2019 [58]	31683883	4T1, MDA-MB-231, MCF-10A	Histone deacetylase inhibitor to induce misfolded proteins	D
Diaz et al., 2013 [63]	24648762	MCF-7	Peroxiredoxin II	I
Fu et al., 2022 [65]	35497337	MDA-MB-231	NSMRH (G2/M)	D
Kumar et al., 2020 [56]	33385162	TP53+/+, Fusion-Reporter (TP53+/+, PCNA-mCherry, 53BP1-mVenus)	DNA-PK- and Pol θ-dependent end joining repair	I
Lamb et al., 2015 [57]	26087309	MCF-7, T47D	DNA-PK	I
Luzhna et al., 2013 [66]	23467667	MCF-7	↑ pATM, KU70, RAD51, and low fidelity DNA polymerase	I
Mehta et al., 2016 [51]	27588488	MDA-MB-231, MDA-MB-468 and Hs578t	HuR	I
Nashir Udden et al., 2023 [54]	36693944	T-47D and MCF-7	BET	I
Nolte et al., 2023 [67]	36835001	MCF-7, MDA-MB-231, BT-20	Microtubule disruption via ESE-16 molecule	D
Park et al., 2017 [68]	28554201	MDA-MB-231	Induction of apoptosis	D
Sencan et al., 2021 [55]	33515382	MDA-MB-231, MDA-MB-436, BT-20, MCF-7, T47D, ZR-75.1, MCF-10A	UVRAG	I
Tian et al., 2024 [49]	38609375	MMTV-PyMT	DTL-RUVBL1/2-β-catenin	I
Wang et al., 2015 [48]	25885449	293 T, HeLa, MCF-7, MDA-MB-231	FEN1	I
Wang et al., 2021 [53]	33813001	MCF7, MDA-MB-231, MDA-MB-468, T47D	FASN	I
Wang et al., 2023 [59]	37485315	4T1 cells in vitro and in BALB/c mice	Cu-doped polypyrrole-based hydrogel injected intratumorally to suppress antioxidant capabilities and increase reactive oxygen species production	I
Wu et al., 2014 [60]	25409124	MCF-7, ZR-75-1, MDA-MB-231	Bcl-2	I
Yang et al., 2020 [69]	33000219	MCF-7, T47D	MV-Edm infection mediated ↓ 53BP1 and ↓ NHEJ	D
Zhou et al., 2020 [70]	32175401	MDA-MB-231	SOD2, CDKN1A	I

Abbreviations: HuR (Human Antigen R), ARID1A (AT-Rich Interactive Domain 1A), FASN (Fatty Acid Synthase), NSMRH (sunitinib nanoparticles @ matrix metalloproteinases-response hydrogel), DNA-PK (DNA-Dependent Protein Kinase), pATM (Phosphorylated ATM), BET (Bromodomain and Extraterminal Domain), UVRAG (UV Radiation Resistance-Associated Gene), FEN1 (Flap Endonuclease 1), SOD2 (Superoxide Dismutase 2), CDKN1A (Cyclin-Dependent Kinase Inhibitor 1A).

### 3.3. Signaling Pathways Associated with Radiation Resistance in Breast Cancer

Signaling pathways regulate intercellular and intracellular communication and influence critical processes such as proliferation, survival, and apoptosis. In breast cancer, aberrant activation of these pathways—often initiated by growth factor binding to transmembrane receptors—leads to persistent downstream signaling through cascades such as PI3K/Akt/mTOR, MAPK/ERK, and Jak/STAT, which are known to promote therapy resistance. These cascades converge on key transcriptional and metabolic programs that enhance tumor cell survival following radiation. Consequently, targeting dysregulated signaling has become a major focus in overcoming radiation resistance in breast cancer. Table 3 summarizes key studies examining signaling targets implicated in radioresistance, and Figure 7 provides a schematic overview of these pathways and their downstream functional effects.

**Table 3 cimb-47-00589-t003:** Summary of studies investigating signaling pathways in breast cancer radiation resistance. This table presents studies that examine molecular targets and interventions affecting radiation resistance through modulation of signaling pathways in breast cancer models. For each study, the authors, year of publication, and PubMed ID (PMID) are provided to identify the source. The model systems used (human breast cancer cell lines or xenografts), molecular targets or interventions, and their observed effect on radiation resistance—either an increase (I) or decrease (D)—are listed. In cases where a defined molecular target was not specified, the experimental strategy or cellular condition investigated is noted.

Authors, Year of Publication	PMID	Model System	Molecular Target/ Intervention	Effect on Radiation Resistance [Increase (I)/Decrease (D)]
Boelens et al., 2014 [71]	25417103	MDA-MB-231, 1833, MDA-436, MDA-157, HCC1937, MDA-468, MCF7, SKBR3, T47D, HCC70	Antiviral/NOTCH3 pathways	I
Braunstein et al., 2008 [72]	18234964	MCF10A, UACC-893, HCC70, BT474	NF-kappaB	I
Brennan et al., 2006 [73]	17085655	Human case study samples, MDA-MB-231, MCF-7, T47D, SKBR3, Hs578T, BT474, HeLa	CA IX	I
Cordes et al., 2003 [74]	14703944	MDA-MB-231	Fibronectin and laminin	I
Heravi et al., 2012 [75]	22357220	4T1	RAF/MEK/ERK/MAP, VEGFR-2, VEGFR-3, PDGFR-β	I
Hu et al., 2016 [76]	27624978	HBL-100, MCF–7, MDA-MB-231, HCC1937, SKBR-3, and BT549	ITGA6	I
Iijima et al., 2018 [77]	29393397	MDA-MB-231	HIF-1α	I
Ji et al., 2023 [78]	37614420	4T-1	Axl	I
Jung et al., 2019 [79]	30893896	T47D, MDA-MB-231, MCF7	TCTP	D
Krautschneider et al., 2022 [80]	36411172	MDA-MB-231, HCC1806	HS chains	I
La Verde et al., 2022 [81]	36091449	MCF10A, MDA-MB-231	YAP	
Lei et al., 2022 [82]	36329030	MCF-12A, MCF-12 F, MCF-7, T47D, ZR-75-1, HCC-1806, MDA-MB-468, BT-549, MDA-MB-231, MCF-10A	lncRNA DUXAP8	I
Li et al., 2021 [83]	34221989	MCF-7R, MDA-MB-231R	LncRNA FGD5-AS1	I
Liang et al., 2022 [84]	35944750	MDA-MB-231, BT- 549, MCF-7, T-47D	CD146, ITGB1	I
Ling et al., 2009 [85]	19956451	MCF-7, ZR-75 and MDA-MB-231	Survivin	I
Luo et al., 2009 [86]	19513620	MCF-7, MDA-MB-453, SK-BR-3	ERBB2	I
Marvaso et al., 2014 [87]	24657936	MDA-MB-361	FTY720	D
Mast & Kuppusamy, 2018 [88]	30524959	MDA-MB-231	Hypoxia	I
Miao et al., 2021 [89]	33739118	MCF-7, MDA-MB-231	TAF9	I
Onaga et al., 2022 [90]	35813014	The Molecular Taxonomy of Breast Cancer International Consortium dataset	SLC20A1	I
Paramanantham et al., 2021 [91]	34066541	MDA-MB-231	ERK	I
Steelman et al., 2011 [92]	21869603	MCF-7	Akt/mTOR	I
Tao et al., 2024 [93]	38167446	MDA-MB-231, MDA-MB-468	PDIA4	I
Thewes et al., 2010 [94]	20459791	N202.1A	AP-2 transcription factors	I
Wolfe et al., 2015 [95]	25832697	SUM 149, KPL4	VLDL; LDL	I for VLDL; D for LDL
Zhou et al., 2018 [62]	29317253	MDA-MB-231	SDF-1 receptor CXCR4	I
Zou et al., 2017 [96]	29169152	BT474, SKBR3, Hs578T and MDA-MB-231	CAVEOLIN-1	I

Abbreviations: NF-kappaB (Nuclear-Factor-kappaB), CA IX (Carbonic Anhydrase IX), RAF (Rapidly Accelerated Fibrosarcoma), MEK (Mitogen-Activated Protein Kinase), ERK (Extracellular Signal-Regulated Kinase), MAP (Mitogen-Activated Protein), VEGFR-2 (Vascular Endothelial Growth Factor 2), VEGFR-3 (Vascular Endothelial Growth Factor 3), PDGFR-β (Platelet-Derived Growth Factor Receptor-Beta), ITGA6 (Integrin Alpha-6), HIF-1α (Hypoxia-Inducible Factor 1-α), TCTP (Translationally Controlled Tumor Protein), HS Chains (Heparan Sulfate Chains), YAP (Yes-associated protein), LncRNA DUXAP8 (Long Non-Coding Ribonucleic Acid Double Homeobox A Pseudogene 8), CD146 (Cluster of Differentiation 146), ITGB1 (Integrin Beta 1), ERBB2 (Erythroblastic Oncogene B-2), FTY720 (Sphingosine Analog Fingolimod), TAF9 (TATA-Box Binding Protein Associated Factor 9), SLC20A1 (Solute carrier family 20 member 1), Akt (Protein Kinase B), mTOR (Mammalian Target of Rapamycin), PDIA4 (Protein Disulfide Isomerase Family Member 4), AP-2 Transcription Factors (Activator protein-2 Transcription Factors), VLDL (very-low-density lipoproteins), LDL (low-density lipoproteins), SDF-1 (stromal cell-derived factor 1), CXCR4 (Chemokine Receptor 4).

#### 3.3.1. PI3K/Akt/mTOR Signaling Pathway

The PI3K/Akt/mTOR pathway is a central intracellular signaling cascade that governs cell proliferation, survival, and growth. Dysregulation of this pathway is commonly observed in therapy-resistant cancers, including breast cancer [97]. Among signaling pathways studied, PI3K/Akt/mTOR was the most frequently targeted in the context of radiation resistance [97]. A few studies chose to evaluate the role of this pathway in radiation resistant breast cancer in targeting specific molecules within the pathway. A study performed by Ji et al. [78] evaluated the connection between Axl, a transmembrane protein, and radiation resistance. Since Axl is known to activate the PI3K/Akt/mTOR pathway, they chose to inhibit it in their radiation resistant induced cell line 4T-1/IRR using R428 [78]. Upon inhibition of Axl, the cells exhibited increased radiation sensitivity [78]. In taking it one step further, they combined R428 therapy and radiation therapy in vivo and found that the combination therapy decreased cell growth in mice more than when the therapies were used alone [78]. Therefore, Axl inhibition may sensitize radiation resistant cells.

Non-coding RNA’s, including long non-coding RNA’s (lncRNA) are known to regulate gene expression and have recently been found to play a role in the malignancy of some cancers [82]. More specifically, the lncRNA DUXAP8 has demonstrated expression in multiple different cancers, such as lungs and bladders. A study performed by Let et al. [82] demonstrated that DUXAP8 was overexpressed in breast cancer and particularly enriched in radioresistant cells. DUXAP8 overexpression suppressed PTEN—a negative regulator of PI3K—and increased phosphorylation of PI3K/Akt/mTOR pathway components. The knockout of DUXAP8 reversed these effects and restored radiation sensitivity. Mechanistically, DUXAP8 also upregulated EZH2, which repressed E-cadherin and RHOB, further promoting resistance. Treatment with a PI3K inhibitor reduced cell viability in DUXAP8-overexpressing cells, supporting its role as a mediator of resistance.

Similarly, Hu et al. [76] examined integrin α6 (ITGA6), an adhesion receptor upregulated in several breast cancer lines. ITGA6 activation promoted phosphorylation of downstream Akt and ERK. Inhibition of PI3K or MEK suppressed these phosphorylation signals and increased radiosensitivity, indicating that ITGA6 enhances resistance through PI3K/Akt and MEK/ERK signaling [76].

Lastly, Marvaso et al. [87] investigated the effects of FTY720, an immunosuppressant and sphingosine analog commonly used in the treatment of multiple sclerosis. Given the role of sphingolipid metabolism in therapy resistance, the authors hypothesized that inhibiting sphingosine kinase 1 (SphK1) with FTY720 could sensitize breast cancer cells to radiation. SphK1 typically converts pro-apoptotic ceramides into sphingosine-1-phosphate (S1P), a lipid signaling molecule that promotes cell survival. When breast cancer cells were treated with 8 Gy radiation in combination with FTY720, levels of phosphorylated ERK1/2 and Akt were markedly reduced compared to radiation alone. This decrease in S1P disrupts downstream activation of survival pathways, thereby enhancing radiosensitivity. Additionally, treatment with FTY720 led to an increase in autophagosome formation, suggesting that both apoptosis and autophagy contributed to the enhanced therapeutic response.

In addition to these mechanisms, Jung et al. [79] examined the role of translationally controlled tumor protein (TCTP), a stress-response protein with anti-apoptotic function. TCTP knockdown in T47D, MDA-MB-231, and MCF-7 cells led to significantly increased radiation-induced cell death, suggesting that TCTP supports radiation resistance by promoting cell survival under genotoxic stress. Given its known interactions with Akt and mTOR pathways in other cancers, TCTP may represent a novel downstream effector of PI3K/Akt-mediated resistance and a potential target for radiosensitization in breast cancer.

Together, these studies underscore the central role of PI3K/Akt/mTOR signaling in modulating radiation resistance and highlight multiple points of therapeutic intervention, including upstream activators, non-coding RNAs, and lipid-mediated signaling modifiers. However, therapeutic targeting of the PI3K/Akt/mTOR axis in isolation may be insufficient due to compensatory activation of parallel signaling cascades such as the MAPK/ERK pathway [98]. For example, PI3K inhibition has been shown to induce ERK phosphorylation, potentially attenuating the radiosensitizing effect [99]. To address this limitation, combination strategies are being explored. Notably, co-inhibition of PI3K and PARP has demonstrated synergistic effects in preclinical breast cancer models, enhancing radiosensitivity by simultaneously disrupting survival signaling and DNA repair [100]. These findings support a multipronged therapeutic approach to overcome adaptive resistance and improve clinical outcomes.

#### 3.3.2. JNK Signaling Pathway

The c-Jun *N*-terminal kinase (JNK) signaling pathway regulates diverse cellular processes including differentiation, proliferation, apoptosis, and stress responses [101]. Dysregulation of this pathway has been implicated in various malignancies and therapy resistance. In the context of breast cancer, Tao et al. [93] investigated the role of the endoplasmic reticulum protein PDIA4 (protein disulfide isomerase family member 4), a member of the protein disulfide isomerase family, in modulating JNK signaling and radiation resistance in TNBC cells. PDIA4 was found to be upregulated in both breast cancer and TNBC cell lines. Mechanistically, PDIA4 interacts with TAX1BP1, a regulatory protein involved in both NF-κB and JNK signaling. PDIA4 binding promoted degradation of TAX1BP1, thereby suppressing JNK activity. Downregulation of PDIA4 restored JNK signaling, leading to increased apoptosis and enhanced sensitivity to radiation. These findings suggest that PDIA4 promotes radiation resistance by inhibiting pro-apoptotic JNK signaling in TNBC [93].

#### 3.3.3. MAPK, MEK/ERK, and Downstream Resistance Pathways

The MAPK/ERK pathway, activated by a variety of extracellular stimuli including growth factors and cytokines, plays a critical role in cell proliferation, survival, and differentiation [102]. In breast cancer, this pathway has been implicated in promoting resistance to radiotherapy through diverse upstream activators. For example, Heravi et al. [75] demonstrated that targeting multiple upstream kinases—including RAF, VEGFR-2/3, and PDGFR-β—reduced radiation resistance in 4T1 breast cancer cells, implicating broad involvement of the MAPK cascade. Similarly, Paramanantham et al. [91] reported that ERK inhibition sensitized MDA-MB-231 cells to radiation, underscoring the therapeutic potential of targeting this pathway directly.

Zhou et al. [62] investigated the chemokine receptor CXCR4, a regulator of SDF-1 signaling that activates ERK. Inhibiting CXCR4 reduced radiation resistance, reinforcing the role of ERK-driven survival signaling. Likewise, Luo et al. [86] further showed that ERBB2 overexpression correlated with increased radiation resistance, partly through MAPK signaling. Finally, other targets associated with MAPK signaling include Survivin, a downstream effector studied by Ling et al. [85], and Notch3, investigated by Boelens et al. [71], both of which were upregulated in resistant cells and contribute to post-radiation survival.

In parallel, La Verde et al. [81] explored the Hippo-YAP/TAZ signaling axis, which influences cell proliferation, apoptosis, and tissue regeneration [103]. Upon exposure to ionizing radiation (2 Gy and 10 Gy), YAP expression and phosphorylation patterns were altered in both healthy and malignant breast epithelial cells. Notably, cancerous cells exhibited a delayed and sustained elevation of YAP levels 72 h post-irradiation, suggesting that YAP activity promotes post-radiation survival and contributes to therapy resistance by enhancing cell proliferation [81].

#### 3.3.4. Hypoxia and Metabolic Signaling

Hypoxia is a well-characterized driver of therapy resistance in tumors, in part by activating hypoxia-inducible factors (HIFs) and altering cellular metabolism [104]. Mast and Kuppusamy [88] demonstrated that hypoxic conditions conferred radiation resistance in MDA-MB-231 cells, a phenomenon attributed to increased survival signaling. Iijima et al. [77] further confirmed that HIF-1α expression was elevated in resistant cells and modulated downstream pro-survival targets. Moreover, Brennan et al. [73] showed that CAIX, a known HIF target, was upregulated in resistant breast cancer models, supporting a mechanistic link between hypoxia and resistance. Additionally, Wolfe et al. [95] investigated the role of lipid metabolism, finding that very low-density lipoprotein (VLDL) exposure increased resistance, while low density lipoprotein (LDL) reduced it, suggesting that metabolic state modulates radiation response through Akt and other survival pathways.

**Figure 7 cimb-47-00589-f007:**
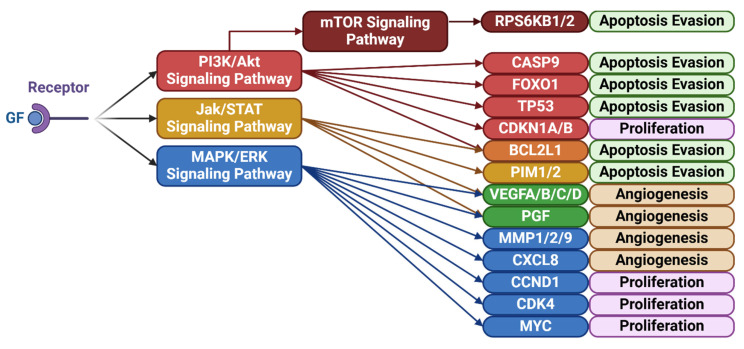
Growth factor–activated signaling pathways and their downstream targets contributing to radiation resistance in breast cancer. Summary of important signaling pathways for cancer progression and some of their downstream effects. The binding of growth factors (GF) such as EGF, PDGF, IGF, HGF, etc., to their respective ligand receptors triggers an effect throughout the PI3K/Akt/mTOR, Jak/STAT, and MAPK/ERK signaling pathways [78]. These pathways influence cancer progression and radiation resistance by influencing the regulation and activity of a variety of proteins which in turn influence factors such as apoptosis evasion, proliferation and angiogenesis [105,106,107]. Created in BioRender. Dickinson, N. (2025) https://BioRender.com/lui2zt2.

### 3.4. MicroRNAs as Post-Transcriptional Regulators of Radiation Resistance in Breast Cancer

MicroRNAs (miRNAs) are short, non-coding RNA molecules that regulate gene expression post-transcriptionally by inhibiting messenger RNA (mRNA) translation or promoting its degradation [108,109]. In cancer, miRNA expression is frequently dysregulated—either through upregulation of oncogenic miRNAs or downregulation of tumor-suppressive miRNAs—contributing to malignancy, therapy resistance, and disease progression. In breast cancer, miRNAs have emerged as critical modulators of radiation response, with several studies demonstrating their potential as biomarkers and therapeutic targets for radiosensitization. Table 4 summarizes studies that have explored the role of specific miRNAs in mediating radiation resistance or sensitivity in breast cancer models.

Yang et al. [69] demonstrated that miRNA-634 expression was significantly reduced in MCF-7 and MDA-MB-231 cells that had been rendered radiation resistant through exposure to a cumulative dose of 50 Gy, delivered in 2 Gy fractions over 11 weeks [69]. To assess the functional role of this downregulation, miRNA-634 mimics were transfected into the radiation-resistant cells. Upon subsequent irradiation, these transfected cells exhibited decreased survival, indicating that reduced miRNA-634 expression may contribute to the development of radiation resistance in breast cancer [69]. The results demonstrated that the survival rate of these cells had decreased upon being exposed to radiation once again, suggesting that low levels of miRNA-634 may lead to radiation-resistant breast cancer cells [69]. Similarly, Wolfe et al. [110] investigated miR-33a and found that its expression was also diminished in radiation-resistant breast cancer cells compared to non-resistant counterparts. Transfection of miR-33a into resistant cells prior to irradiation restored radiosensitivity, suggesting a protective role for this miRNA in modulating radiation response.

In contrast, some miRNAs were upregulated in resistant cells and promoted survival. Anastasov et al. [111] reported that miR-21 expression significantly increased in T47D invasive ductal carcinoma cells following irradiation. To test whether miR-21 contributes to radiation resistance, the researchers pre-treated cells with an anti-miR-21 inhibitor prior to radiation exposure. This intervention led to a 20% increase in apoptosis compared to untreated controls, suggesting that elevated miR-21 levels play a role in protecting cells from radiation-induced cell death. In a separate study, Yu et al. [112] identified miR-144 as another miRNA upregulated in radiation-resistant breast cancer cells. Using MDA-MB-231 and HER2-positive SKBR3 cell lines, they demonstrated that inhibition of miR-144 enhanced radiation-induced apoptosis, whereas overexpression of miR-144 promoted resistance. Mechanistically, miR-144 was shown to facilitate EMT, increase cellular migration and invasion, and modulate key signaling proteins including AKT and PTEN—further linking this miRNA to aggressive, therapy-resistant phenotypes [112].

Collectively, these studies highlight the dual roles of miRNAs in either promoting or suppressing radiation resistance in breast cancer, depending on their expression patterns and molecular targets. Targeting oncogenic miRNAs or restoring tumor-suppressive miRNAs may offer novel strategies to enhance radiosensitivity. These mechanisms are visually summarized in Figure 8, which contrasts miRNA expression profiles in radiation-sensitive versus radiation-resistant breast cancer cells.

**Table 4 cimb-47-00589-t004:** Summary of studies investigating microRNAs in breast cancer radiation resistance. This table summarizes studies examining the role of microRNAs (miRNAs) in modulating radiation resistance in breast cancer models. For each study, the authors, year of publication, and PubMed ID (PMID) are provided to identify the source. The model systems used (human breast cancer cell lines or xenografts), molecular targets or interventions, and their observed effect on radiation resistance—either an increase (I) or decrease (D)—are listed. In cases where a defined molecular target was not specified, the experimental strategy or cellular condition investigated is noted.

Authors, Year of Publication	PMID	Model System	Molecular Target/ Intervention	Effect on Radiation Resistance [Increase (I)/Decrease (D)]
Anastasov et al., 2012 [111]	23216894	T47D, MDA-MB-361	miR-21	I
Masoudi-Khoram et al., 2020 [113]	32493932	MDA-MB-231, T47D	miR-16-5p	D
Mesci et al., 2017 [114]	28419078	MDA-MB-231	miR-330-3p and CCBE1	I
Wang et al., 2022 [115]	35818245	human breast cancer cells (specific not mentioned)	miR-143-3p (through FGF1)	D
Wolfe et al., 2016 [110]	27055396	SUM149, SUM159, KPL4, MDA-MB-231	miR-33a	I
Yang et al., 2020 [69]	32077744	MCF-7, MDA-MB-231	miR-634, STAT3	D
Yu et al., 2015 [112]	26252024	MDA-MB-231, SKBR3	miR-144	I
Zhang et al., 2020 [116]	32374522	MCF-7, T47D, LM-MCF-7,34 BT-474, SKBR-3, MDA-MB-231	miR-449b-5p	D

Abbreviations: miR (microRNA), CCBE1 (Collagen and Calcium Binding EGF Domains 1), FGF1 (Fibroblast growth factor-1), STAT3 (Signal Transducer and Activator of Transcription 3).

### 3.5. Novel and Less-Explored Molecular Targets in Radiation Resistance

In addition to well-characterized signaling, DNA repair, and immune pathways, a range of novel molecular interventions have been investigated for their potential to modulate radiation resistance in breast cancer. These include nanoparticle-based therapies, immune modulation, and stromal cell interactions. While these mechanisms are less frequently studied, they offer promising alternative strategies for radiosensitization. Table 5 summarizes these studies and their effects on radiation response.

#### 3.5.1. Nanoparticle-Based Approaches

Jain et al. [117] evaluated the use of gold nanoparticles under varying oxygen conditions and found that these particles enhanced radiosensitivity of MDA-MB-231 cells in normoxic (21% O_2_) and moderately hypoxic (1% O_2_) conditions. However, this radiosensitization was ineffective under extreme hypoxia (0.1% O_2_), likely due to impaired ROS formation, a key mechanism of nanoparticle-induced cell damage.

Zuo et al. [118] extended this approach by engineering gold nanoparticles conjugated with thiolate cupferron, a heat-sensitive nitric oxide (NO) donor. When combined with near-infrared (NIR) laser and x-ray irradiation, this nanoplatform triggered the release of NO, ROS, and reactive nitrogen species (RNS), producing potent radiosensitization both in vitro and in vivo. This multi-modal approach exemplifies the potential of nanoparticle-enhanced radiotherapy in breast cancer (Figure 9).

**Figure 9 cimb-47-00589-f009:**
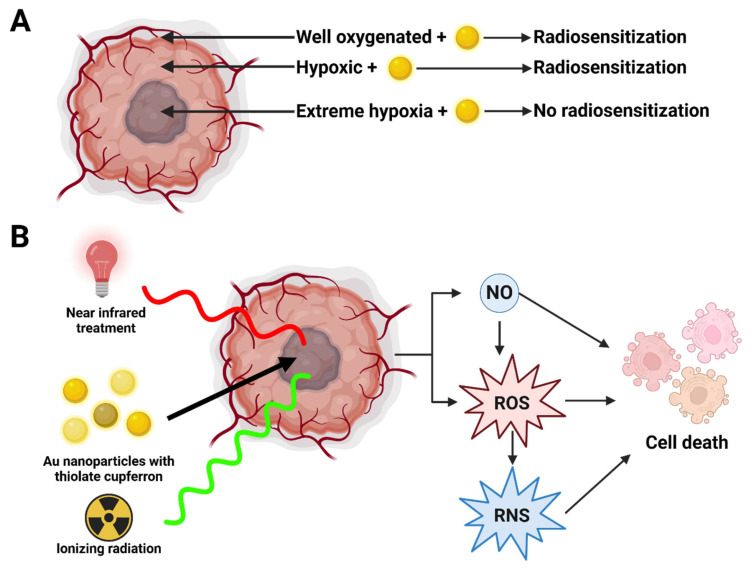
Nanoparticle-based strategies to overcome hypoxia-driven radiation resistance in breast cancer. (**A**) The outermost portion of a tumor is typically vascularized and well oxygenated, while the deeper portions of the tumor are less oxygenated [117]. Gold nanoparticles can increase the radiosensitivity of the cancerous cells although their effectiveness relies on the formation of ROS, so extreme hypoxia negates the change in radiosensitivity caused by the gold nanoparticles [117]. (**B**) Breast cancer when treated with gold nanoparticles loaded with thiolate cupferron, near-infrared treatment and ionizing radiation results in the formation of nitric oxide (NO), ROS, and reactive nitrogen species (RNS) causing cell death [118]. Created in BioRender. Dickinson, N. (2025) https://BioRender.com/e9c8tmv.

In addition, Hullo et al. [119] explored platinum-based nanoparticles in T47D and MDA-MB-231 cells, although the specific effects on radiation response were not reported. Nonetheless, the study highlights ongoing interest in metal-based nanotherapeutics as emerging radiosensitizers.

#### 3.5.2. Immune Modulation and Stromal Targets

The tumor microenvironment plays a critical role in shaping the response of breast cancer cells to radiation [120,121]. In particular, components of the immune system and stromal cell populations such as cancer-associated fibroblasts (CAFs) can either support or suppress tumor survival following irradiation. Radiation has been shown to alter immune recognition, modulate cytokine production, and affect immune cell infiltration within tumors [122]. At the same time, stromal elements—including fibroblasts, extracellular matrix components, and secreted factors—can confer protective signals that facilitate tumor regrowth and resistance [123,124,125]. Targeting these non-tumor cell populations offers a promising avenue to improve the efficacy of radiotherapy by disrupting the supportive interactions that sustain resistant phenotypes.

Choi et al. [126] investigated how mebendazole—a repurposed antiparasitic agent—could enhance natural killer (NK) cell-mediated cytotoxicity when used in combination with radiation. The dual treatment improved immune clearance of MDA-MB-231 cells, demonstrating that modulating immune cell function may provide synergistic benefits with radiotherapy.

Jian et al. [127] focused on cancer-associated fibroblasts (CAFs), a key component of the tumor microenvironment. CAFs were shown to promote radiation resistance in 4T1 breast cancer cells, likely through paracrine signaling and extracellular matrix remodeling, which support tumor survival and recovery post-irradiation.

**Table 5 cimb-47-00589-t005:** Summary of studies investigating novel or less-explored molecular targets in breast cancer radiation resistance. This table compiles studies that examine the effect of emerging or unconventional targets—such as nanoparticles, immune modulation, or stromal interactions—on radiation resistance in breast cancer models. For each study, the authors, year of publication, and PubMed ID (PMID) are provided to identify the source. The model systems used (human breast cancer cell lines or xenografts), molecular targets or interventions, and their observed effect on radiation resistance—either an increase (I) or decrease (D)—are listed. In cases where a defined molecular target was not specified, the experimental strategy or cellular condition investigated is noted.

Authors, Year of Publication	PMID	Model System	Molecular Target/ Intervention	Effect on Radiation Resistance [Increase (I)/Decrease (D)]
Choi et al., 2022 [126]	36555137	MDA-MB-231	Natural killer cell mediated cytotoxicity increased following mebendazole and radiation	D
Hullo et al., 2021 [119]	33922713	T47D, MDA-MB-231	Platinum nanoparticles	N/A
Jain et al., 2014 [117]	24444528	MDA-MB-231	Gold nanoparticles	D
Jian et al., 2024 [127]	38214439	4T1	CAFs	I
Zuo et al., 2023 [118]	36686245	MCF-7, MCF-10A	Gold nanoparticles	D

Abbreviations: CAFs (cancer-associated fibroblasts).

## 4. Summary and Future Directions

This review provided an in-depth evaluation of the molecular determinants and biological processes implicated in radiation resistance in breast cancer. Across the literature, the most extensively studied mechanisms centered on alterations in DNA damage response pathways and survival signaling cascades. For example, the PI3K/AKT/mTOR axis emerged as a recurrent target, with studies demonstrating that pharmacologic or genetic inhibition of key transmembrane receptors such as Axl and ITGA6 could reverse radiation resistance. Similarly, several studies focused on targeting DNA repair regulators within the NHEJ pathway, including ARID1A and RUVBL1, highlighting their roles in facilitating DNA double-strand break repair and maintaining radioresistant phenotypes. Some research also extended into therapeutic repurposing, where drugs like proton pump inhibitors or selective estrogen receptor modulators sensitized breast cancer cells to ionizing radiation.

Radiation resistance also appears closely linked to the presence of breast cancer stem-like cells marked by CD44(+)/CD24(−/low) expression. These subpopulations were shown to activate mTOR and ATM pathways and express high levels of ALDH1, providing multiple potential radiosensitization targets. Additionally, miRNAs were frequently implicated as key regulators of radiation response. Both gain- and loss-of-function strategies involving miRNAs such as miR-634, miR-33a, miR-21, and miR-144 demonstrated that modulation of post-transcriptional control could substantially affect cell survival post-irradiation.

Despite these advances, key limitations were evident across the body of literature. The majority of studies were confined to in vitro models and disproportionately relied on a limited number of breast cancer cell lines—most notably MCF-7 and MDA-MB-231—raising concerns about the generalizability of findings to the broader spectrum of breast cancer phenotypes. This narrow representation may overlook subtype-specific mechanisms or interactions relevant to the tumor microenvironment. Additionally, much of the research has naturally concentrated on well-established signaling pathways—so-called “usual suspects”—such as PI3K/Akt/mTOR or DNA repair pathways, due to their well-characterized roles in cancer biology. While this targeted focus has yielded valuable insights, many studies examined these pathways in isolation, limiting the discovery of broader network-level interactions or alternative resistance mechanisms. Compounding this, inconsistent molecular targets and variable experimental methodologies have led to challenges in synthesizing a unified model of radiation resistance in breast cancer.

Furthermore, there is a critical lack of in vivo validation across the literature. Most studies rely on established cell lines under artificial 2D conditions, which fail to recapitulate the spatial, genetic, and phenotypic heterogeneity observed in tumors. Discrepancies between cell line data and clinical outcomes—particularly with respect to hypoxia-induced signaling and stromal interactions—underscore the need for more physiologically relevant models. Future research should incorporate patient-derived xenografts (PDXs), organoid cultures, and genetically engineered mouse models to better reflect tumor–microenvironment dynamics and radiation response. These platforms will be essential to evaluate the translational relevance of proposed radiosensitization strategies and to identify resistance mechanisms driven by cell–stromal or immune interactions.

To advance our understanding of radiation resistance in breast cancer, future research should prioritize the use of unbiased, genome-wide functional screening approaches. Techniques such as CRISPR-Cas9-mediated gene knockout or activation libraries applied to clinically relevant, therapy-resistant breast cancer models hold great promise for identifying previously unrecognized genes and pathways that contribute to treatment failure. These high-throughput screening strategies can systematically map genetic dependencies and highlight novel molecular targets that are not apparent through candidate-based studies alone.

In parallel, integrative multi-omics approaches—including transcriptomics, proteomics, and epigenomics—are essential for constructing a comprehensive view of the regulatory networks driving radiation resistance. By combining these datasets, researchers can capture not only gene expression changes but also post-translational modifications, chromatin remodeling, and epigenetic alterations that influence cellular responses to radiation. This systems-level perspective will be instrumental in identifying patient-specific vulnerabilities and in guiding the development of personalized radiosensitization strategies.

Importantly, the tumor microenvironment remains underexplored. Cancer-associated fibroblasts, extracellular matrix remodeling, and immune cell interactions significantly influence tumor adaptation to radiation yet were only sparsely covered in current studies. Enhanced focus on stromal and immune components—such as NK cell function and stromal-mediated paracrine signaling—could open new avenues for combination therapies that disrupt both cancer-intrinsic and extrinsic drivers of resistance.

Given the multiplicity of resistance mechanisms, future strategies should emphasize combination therapies that target multiple molecular pathways in parallel. This multi-pronged approach—akin to combination chemotherapy—may offer a more robust means of sensitizing resistant tumor populations to radiation.

## 5. Conclusions

In conclusion, while considerable progress has been made in delineating the molecular contributors to radiation resistance in breast cancer, significant challenges remain. Up to 20–30% of breast cancer patients experience local or distant recurrence, often linked to therapy-resistant tumor subpopulations that survive standard treatment [128]. Given that radiation therapy is a cornerstone of breast cancer management, overcoming resistance is crucial to improving long-term outcomes. Future efforts must therefore prioritize in vivo validation, multi-omics integration, and tumor ecosystem modeling, including immune and stromal components, to ensure translational relevance. By addressing these gaps, research can more effectively guide the development of personalized radiosensitization strategies and reduce recurrence rates in this substantial patient population.

## Figures and Tables

**Figure 1 cimb-47-00589-f001:**
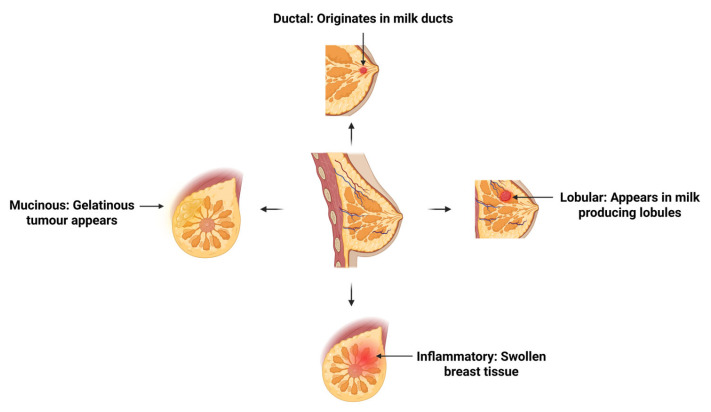
Histological subtypes of breast cancer. Illustration of four common histological subtypes of breast cancer: ductal, lobular, mucinous, and inflammatory. Ductal and lobular carcinomas originate from the milk ducts and lobules, respectively. Mucinous carcinoma is characterized by tumor cells suspended in mucin pools, while inflammatory breast cancer presents with dermal lymphatic invasion, causing erythema and edema. Created with BioRender. Mageau, E. (2025) https://BioRender.com/vp8dakl.

**Figure 2 cimb-47-00589-f002:**
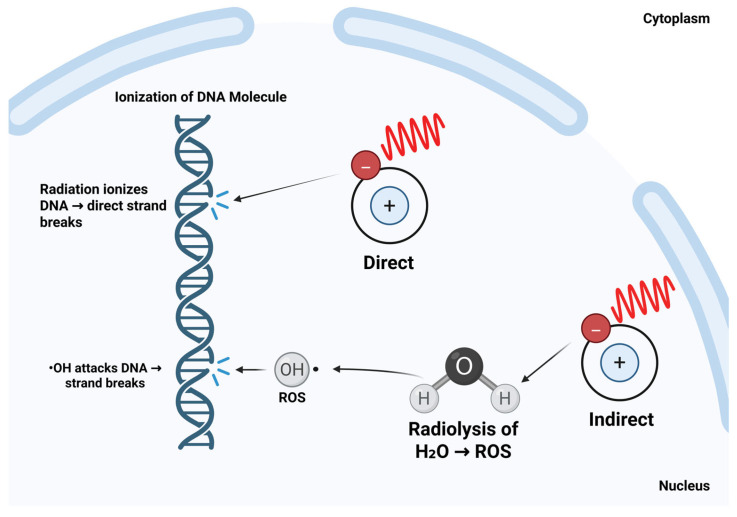
Mechanisms of radiation-induced DNA damage: direct and indirect effects. Ionizing radiation damages DNA through two primary mechanisms. In direct damage, radiation interacts directly with DNA molecules, causing strand breaks and base modifications. In indirect damage, radiation interacts with cellular water to generate reactive oxygen species (ROS), which then chemically damage DNA. Both pathways contribute to the cytotoxic effects of radiation therapy. Created in BioRender. Derbowka, R. (2025) https://BioRender.com/49mv0nd.

**Figure 3 cimb-47-00589-f003:**
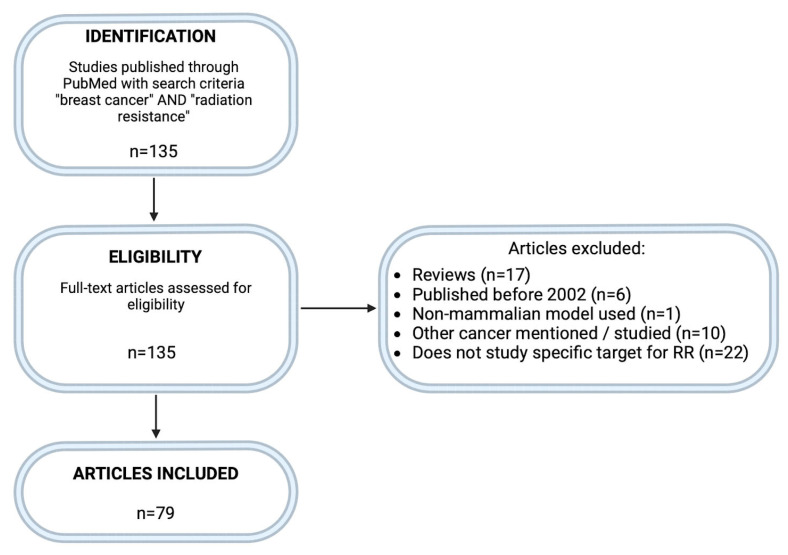
PRISMA-based flow diagram of study selection for systematic review. Summary of the identification, screening, and inclusion process for studies selected in this systematic review. A total of 135 articles were identified through PubMed using the search terms “breast cancer” AND “radiation resistance.” After applying exclusion criteria—reviews, pre-2002 publications, non-mammalian models, non-breast cancers, and studies lacking specific molecular targets—79 articles remained for inclusion. Created in BioRender. Mageau, E. (2025) https://BioRender.com/zgehzac.

**Figure 4 cimb-47-00589-f004:**
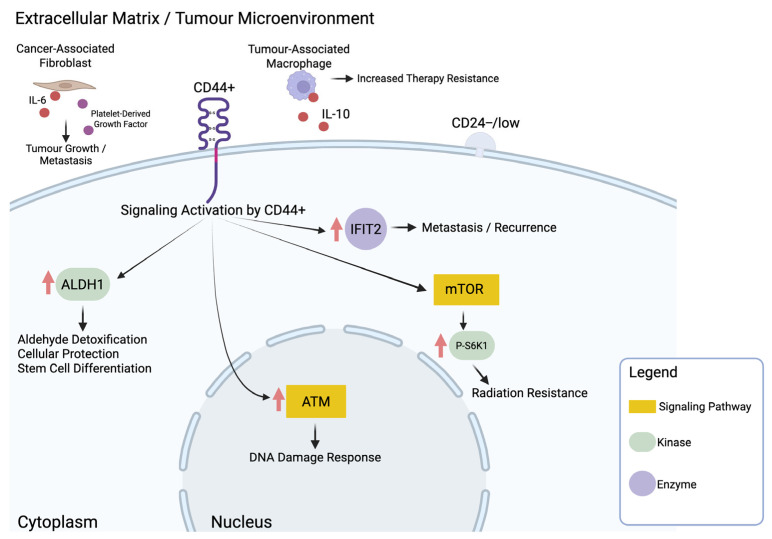
Key molecular pathways activated in CD44^+^/CD24^−^/low breast cancer stem cells contributing to radiation resistance. Schematic representation of molecular targets and signaling pathways associated with radiation resistance in breast cancer stem cells exhibiting the CD44^+^/CD24^−^/low phenotype. CD44-mediated signaling has been linked to the upregulation of ALDH1, IFIT2, ATM, and the mTOR pathway, including its downstream effector phosphorylated S6 kinase 1 (p-S6K1). These factors contribute to enhanced DNA repair capacity, survival, and resistance to radiotherapy. The tumor microenvironment further enhances these characteristics in breast cancer through the secretion of interleukin-6 (IL-6) and platelet-derived growth factors from cancer-associated fibroblasts. Tumor-associated macrophages further contribute to therapy resistance through the secretion of interleukin-10 (IL-10). Created in BioRender. Mageau, E. (2025) https://BioRender.com/c6b46ka.

**Figure 5 cimb-47-00589-f005:**
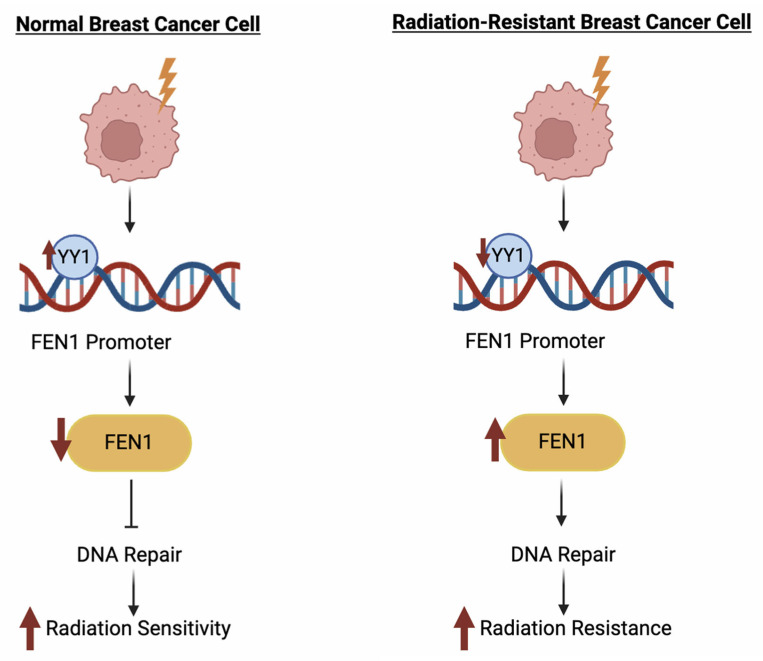
Regulation of FEN1 by YY1 and its impact on radiation sensitivity in breast cancer cells. This figure illustrates the proposed mechanism by which the transcription factor YY1 regulates FEN1 expression and influences DNA repair capacity and radiation response. In normal breast cancer cells, YY1 binds to the FEN1 promoter and suppresses its expression, resulting in reduced DNA repair activity and increased radiation sensitivity. In contrast, radiation-resistant breast cancer cells exhibit reduced YY1 expression, leading to upregulation of FEN1, enhanced DNA repair, and increased radiation resistance. Created in BioRender. Mageau, E. (2025) https://BioRender.com/vtbz5js.

**Figure 8 cimb-47-00589-f008:**
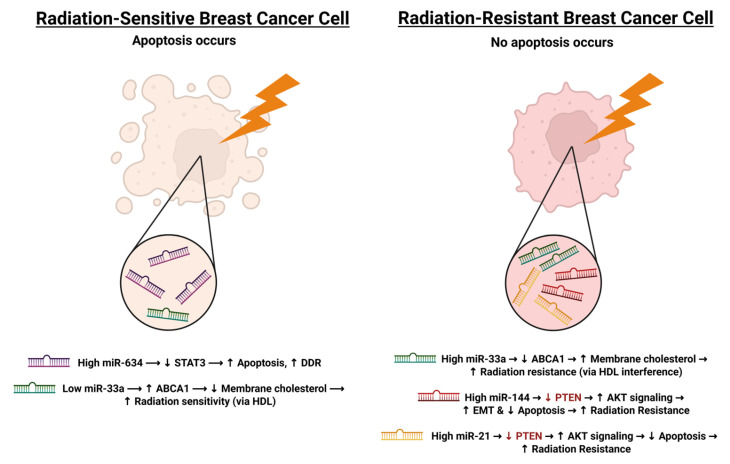
Differential expression of key miRNAs in radiation-sensitive and radiation-resistant breast cancer cells. This figure illustrates the contrasting expression profiles of miRNAs in radiation-sensitive versus radiation-resistant breast cancer cells following ionizing radiation. In radiation-sensitive cells, high miR-634 expression suppresses STAT3, leading to increased rates of apoptosis and DDR. In contrast, lower levels of miR-33a allows for higher expression of ABCA1, facilitating membrane cholesterol efflux via HDL and increasing radiosensitivity. In contrast, radiation-resistant cells exhibit higher levels of miR-33a, leading to higher radiation resistance. miR-144 and miR-21 are both involved in downregulation of PTEN, leading to increased AKT signaling and resulting in increased radiation resistance. Red arrows and text indicates downregulation. Created in BioRender. Derbowka, R. (2025) https://BioRender.com/ricemzw.

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
