# Peer review of "Molecular Mechanisms of Radiation Resistance in Breast Cancer: A Systematic Review of Radiosensitization Strategies"

_cimb, 2025, doi:10.3390/cimb47080589_

Round 1

Reviewer 1 Report

Comments and Suggestions for Authors

The authors did a thorough job of compiling papers over more than 20 years that seek to study radiation resistance in breast cancer.  The research is broken down in different topics and analyzed relative to the topic and findings.  I believe the analysis of the papers and research performed are well explained and analyzed.  I found the review to be informative and thorough.  I have only a few comments that I think would help create a stronger narrative and presentation of data.

  1. In the final version, I would try to keep all tables to a single page.  I think inserting a line below each individual reference in the table would make it easier to read.
  2. While I appreciate the use of figures to enhance the written content, I did not think that most of them were necessary.  I appreciated Figures 3 and 7 but think that the other illustrations did not add to the content of the review.  Instead, I would have liked to see a figure that incorporates all the different mechanisms by which radiation resistance can occur – how they function under normal circumstances and how upregulation or downregulation causes increased radiation resistance.
  3. In the conclusion, I think it would be valuable to mention that while the research and understanding in the field has demonstrated growth and understanding, due to the large number of ways that radiation resistance may occur, future directions should focus on targeting and understanding multiple pathways and how they work together.  Just like with chemotherapy and the use of multiple agents, perhaps targeting multiple pathways would result in creating greater radiation sensitivity.

Reviewer 2 Report

Comments and Suggestions for Authors

This systematic review provides a comprehensive synthesis of current knowledge on radiation resistance mechanisms in breast cancer and potential radiosensitization strategies. The manuscript is well-structured, methodologically rigorous, and addresses a clinically significant issue. However, several areas could benefit from refinement to enhance clarity, translational relevance, and scientific impact.

  1. Please add other ‘in vivo’ data, for example, patient-derived xenografts or clinical studies. Include a subsection discussing discrepancies between cell lines and clinical observations, particularly regarding hypoxia and stromal interactions.

  1. While PI3K/AKT/mTOR and DNA repair pathways are well-reviewed, the manuscript could better address compensatory mechanisms (e.g., cross-talk between PI3K and MAPK pathways) that may limit targeted therapies. Add data about the combination therapy (e.g., PI3K inhibitors + PARP inhibitors) to overcome resistance if any.

  1. The miRNA section (e.g., miR-21, miR-144) is informative but lacks mechanistic depth. How do these miRNAs interact with canonical pathways like EMT or DDR? Include a diagram mapping miRNA-mRNA networks to illustrate regulatory hubs.

  1. The TME’s role in radiation resistance is underexplored. CAFs and immune cells (e.g., NK cells) are briefly mentioned but warrant deeper discussion. Add a dedicated section on TME-driven resistance, including extracellular matrix remodeling and immune evasion post-radiation.

  1. Many targets (e.g., FASN inhibitors, gold nanoparticles) lack FDA-approved agents due to toxicity or bioavailability issues. The review could better delineate barriers to clinical adoption.Include a table ranking targets by clinical feasibility (e.g., phase of development, toxicity profiles).

  1. Figure 4 (CSC pathways) is excellent but could incorporate cross-talk with the TME. Table 1 should clarify if "Increase (I)/Decrease (D)" refers to resistance or sensitivity (currently ambiguous).

Reviewer 3 Report

Comments and Suggestions for Authors

  1. Figures 1 to 5 should be redesigned to present more scientific clarity and visual impact. Including annotations, proper legends, and detailed schematics will enhance comprehension and quality.
  2. The current plagiarism or similarity index is above the acceptable threshold. Please ensure it is reduced to less than 15% by rewriting or paraphrasing overlapping content.
  3. The introduction lacks depth and logical flow. It is recommended to rewrite it as a cohesive and continuous paragraph, providing a stronger background, rationale, and objective of the manuscript.
  4. Multiple sections of the manuscript are fragmented into several small paragraphs. Please consolidate related ideas into well-structured paragraphs to improve readability and academic tone.
  5. Please revise all tables to ensure that references are clearly cited at the end of each corresponding row.
  6. Authors are encouraged to include high-quality reprint images (with proper permission) from key referenced articles to support and enrich the discussion, especially in a review manuscript.
  7. The manuscript currently is structured similar to a research article (e.g., Intro, Methodology, Results). However, if this is intended to be a review article, it must be restructured accordingly, with thematic subheadings
  8. The manuscript requires a thorough language and grammar check. Multiple minor errors are found throughout the text, which affect clarity and professionalism.
  9. The conclusion is excessively long and should be condensed to focus on key takeaways, future perspectives, and concise remarks.

Comments on the Quality of English Language

The English could be improved to more clearly express the research.

Round 2

Reviewer 3 Report

Comments and Suggestions for Authors

Accept in its present form